# Renal Thrombotic Microangiopathy in Concurrent COVID-19 Vaccination and Infection

**DOI:** 10.3390/pathogens10081045

**Published:** 2021-08-17

**Authors:** Marco De Fabritiis, Maria Laura Angelini, Benedetta Fabbrizio, Giovanna Cenacchi, Claudio Americo, Stefania Cristino, Maria Francesca Lifrieri, Maria Cappuccilli, Alessandra Spazzoli, Loretta Zambianchi, Giovanni Mosconi

**Affiliations:** 1Nephrology and Dialysis Unit, Morgagni-Pierantoni Hospital, AUSL Romagna, 47121 Forlì, Italy; marco.defabritiis@auslromagna.it (M.D.F.); marialaura.angelini@auslromagna.it (M.L.A.); claudio.americo@auslromagna.it (C.A.); stefania.cristino@auslromagna.it (S.C.); mariafrancesca.lifrieri@auslromagna.it (M.F.L.); alessandra.spazzoli@auslromagna.it (A.S.); loretta.zambianchi@auslromagna.it (L.Z.); 2Pathology Unit, IRCCS-Azienda Ospedaliero-Universitaria di Bologna, 40138 Bologna, Italy; benedetta.fabbrizio@aosp.bo.it; 3Department of Biomedical and Neuromotor Sciences, Alma Mater Studiorum University of Bologna, 40138 Bologna, Italy; giovanna.cenacchi@unibo.it; 4Pathological Anatomy, Sector of Diagnostic and Subcellular Pathology, IRCCS-Azienda Ospedaliero-Universitaria di Bologna, Alma Mater Studiorum University of Bologna, 40138 Bologna, Italy; 5Nephrology, Dialysis and Renal Transplant Unit, IRCCS-Azienda Ospedaliero-Universitaria di Bologna, Alma Mater Studiorum University of Bologna, 40138 Bologna, Italy; maria.cappuccilli@unibo.it

**Keywords:** COVID-19 vaccination, endothelial injury, kidney disease, SARS-CoV-2 infection, thrombotic microangiopathy

## Abstract

We report on the development of nephrotic proteinuria and microhematuria, with histological features of renal thrombotic microangiopathy (TMA), following the first dose of BNT162b2 COVID-19 vaccine (Pfizer-BioNTech) and COVID-19 diagnosis. A 35-year-old previously healthy man was admitted at our hospital due to the onset of foamy urine. Previously, 40 days earlier, he had received the first injection of the vaccine, and 33 days earlier, the RT-PCR for SARS-CoV-2 tested positive. Laboratory tests showed nephrotic proteinuria (7.9 gr/day), microhematuria, serum creatinine 0.91 mg/dL. Kidney biopsy revealed ultrastructural evidence of severe endothelial cell injury suggestive of a starting phase of TMA. After high-dose steroid treatment administration, complete remission of proteinuria was achieved in a few weeks. The association of COVID-19 with renal TMA has been previously described only in patients with acute renal injury. Besides, the correlation with COVID-19 vaccine has not been reported so far. The close temporal proximity (7 days) between the two events opens the question whether the histological findings should be ascribed to COVID-19 itself or to vaccine injection.

## 1. Introduction

Until the end of June 2021, more than 182 million confirmed SARS-CoV-2 infections had been reported in over 200 different countries with 3.9 million deaths. Up to now, about 4.21 billion doses of vaccine have been administered worldwide. The potential adverse effects of COVID-19 vaccines represent a hot topic of current research. Major side-effects appear to be uncommon, although some cases of new onset or relapsing minimal change disease [1,2,3], IgA nephropathy and anti-GBM glomerulonephritis [4] and relapse of IgG4-related disease [5] have recently reported following the Pfizer-BioNTech vaccine. On the other hand, the kidney is one of the main targets of COVID-19 complications related to several mechanisms, including viral cytopathic effects in renal cells, host hyperinflammatory response and endothelial dysfunction. Acute endothelial cell injury and thrombotic microangiopathy have been reported in SARS-CoV-2–infected patients with concurrent acute kidney injury (AKI) or proteinuria [6].

We describe here a case of nephrotic proteinuria and microhematuria, with histological features of renal thrombotic microangiopathy (TMA), without AKI, starting 40 days after the first injection of the BNT162b2 COVID-19 vaccine (Pfizer-BioNTech) and 33 days after SARS-CoV-2 positivity detection through PCR test.

## 2. Case Report

This is a case report of renal TMA following COVID-19 infection and the first dose of Pfizer-BioNTech vaccination. A written informed consent was obtained from the patient to publish this paper. Approval from local ethics committee was waived, as it is not required at our institution for a single case report.

A 35-year-old previously healthy man was admitted at our nephrology center following the onset of foamy urine. Previously, 40 days earlier, he had received the first injection of Pfizer-BioNTech vaccine and 33 days earlier the RT-PCR test for SARS-CoV-2, required for the presence of anosmia, was positive. He described moderate fatigue the day after vaccination and anosmia the day before SARS-CoV-2 test. During the first 3 weeks, he reported intermittent intake of paracetamol. The subsequent clinical symptoms were dysgeusia, myalgia, pharyngodynia. It is also possible that proteinuria was already present before the patient noticed foamy urine. The timeline of clinical events is shown in Figure 1. 

Blood pressure and heart rate were regular. Physical examination did not reveal edema. Biochemistry and hematology tests were within the normal range: serum creatinine 0.91 mg/dL, albumin 4.5 g/L, cholesterol 170 mg/dL, triglycerides 77 mg/dL, hemoglobin 14.5 g/dL, white blood cell count 4.35 × 10^3^/µL, platelets 226 × 10^3^/µL. HBsAg, HCV and HIV Ab were negative, C3 and C4 levels were also unaltered, and ANCA, anti-GBM Ab, ANA and anti-PLA2R Ab were negative. PCR test for COVID-19 was negative. The antibody titer of IgG against spike protein subunits S1 and S2 (S1/S2-IgG), measured using an indirect chemiluminescence immunoassay (LIAISON® SARS-CoV-2 S1/S2 IgG, DiaSorin, Saluggia, Italy), was 62 AU/mL. Spot urinalysis revealed a moderate presence of proteins (300 mg/dL). Urinary sediment showed 476 red blood cells PHPF, 24-hr urinary protein excretion was 7.9 grams. At renal ultrasound, kidneys appeared of normal size with no evidence of urinary tract obstruction. 

Percutaneous kidney biopsy detected 44 glomeruli: at light microscopy (Figure 2A,B), 2 of them showed global sclerosis, while the others were histologically unremarkable with some non-specific features, such as slight expansion of mesangial matrix, podocyte hypertrophy, focal tubular atrophy/interstitial fibrosis, focal small hemolyzed red cell cast and mild arteriolar hyalinosis Immunofluorescence was negative. Three glomeruli were extensively examined by electron microscopy (Figure 2C,D) that highlighted severe endothelial cell injury, with an ultrastructural evidence of mild inner lamina rara widening and loss of endothelial cell fenestrae, occlusion of glomerular capillaries by swollen endothelial cells, podocyte hypertrophy with foot processes regularly aligned and interdigitated. A platelet was also detectable in the vessel lumen. Fibrin precipitates were not detected. 

In summary, light microscopy and immunofluorescence seemed to be consistent with minimal change disease, but electron microscopy excluded this possibility, rather showing some ultrastructural alterations suggestive of an initial phase of renal TMA. Other renal diseases, such as focal and segmental glomerulosclerosis, post-infectious glomerulonephritis, membranoproliferative glomerulonephritis, have been described to trigger endothelial swelling and lamina rara widening, but the histological features of light microscopy and the specific viral and bacterial tests ruled out these hypotheses. 

High-dose steroid treatment was initiated with 1 gram methylprednisolone for 3 days, followed by prednisone 60 mg/day and then reduction to 10 mg every 4 weeks. The patient was discharged after nine days of hospitalization. A total of 2 weeks later, proteinuria decreased to 0.8 grams/day with microhematuria still present, and then in the following 4 weeks, complete remission of proteinuria (0.07 gr/day) and microhematuria was achieved.

## 3. Discussion

To the best of our knowledge, we describe here the first case of urine abnormalities with histological features of initial renal TMA without renal failure, showing an interesting temporal correlation with both positive PCR test for COVID-19 of mild severity and first dose of BNT162b2 COVID-19 vaccine (Pfizer-BioNTech). 

Several indirect pathways of COVID-related kidney injury have been suggested, mainly attributed to the combined effects of virus-induced cytokine storm with complement and coagulation cascades [7,8]. SARS-CoV-2 can trigger endothelial dysfunction by some specific mechanisms: the virus directly infects endothelial cells owing to their high expression levels of angiotensin-converting enzyme 2 (ACE2) and also activates the complement system. In addition, complexes of COVID-19 specific antibodies and viral antigens might elicit endothelial cell injury through the activation of the C1 complex of the classical pathway and the induction of antibody-dependent cytotoxicity. Pro-inflammatory cytokines released by activated macrophages amplify the vicious cycle of vascular integrity disruption, vessel coagulation and thrombosis by degrading the endothelial glycocalyx, activating the coagulation system and dampening anticoagulant mechanisms. The adhesive phenotype of endothelial cells induced by inflammatory cytokines promotes neutrophil infiltration with subsequent massive release of histotoxic mediators, ultimately leading to injury of endothelial cells [9].

In previous reports, COVID-19 patients who underwent kidney biopsy showed heterogeneous kidney histologic features, including acute tubular injury, collapsing glomerulopathy, focal segmental glomerulosclerosis, tubulointerstitial nephritis, thrombotic microangiopathy [6,10,11]. Acute endothelial cell injury and podocytopathy were common histological findings in the setting of AKI, associated with collapsing glomerulopathy, and also in kidney transplant antibody-mediated rejection [12]. Our patient, instead, presented a normal native kidney function and no pre-existing conditions.

In a retrospective study of 110 hospitalized COVID-19 patients, the urinalysis data of proteinuria and hematuria were strongly associated with the development of AKI [9]. In our report, the presence of podocyte hypertrophy, endothelial swelling, without AKI and clinical signs of TMA, such as anemia and thrombocytopenia, can be interpreted as an early detection of renal TMA or a mild expression of COVID-related kidney disease. The onset of these pathological findings, indeed, was chronologically close not only to COVID-19 diagnosis, but also to the first dose of BNT162b2 COVID-19 vaccine administrated one week before SARS-CoV-2 positivity detection. 

Up to now, the association of COVID-19 with renal TMA has been described only in patients with AKI [2,6]. However, no correlation with COVID-19 vaccine has been reported so far. The pathogenesis of vaccine-associated glomerular lesions is fully understood for mRNA-based vaccines able to temporarily induce the host cell to produce the SARS-CoV-2 spike protein. B- and T-lymphocytes are involved in effective immune response to the spike protein. On the other hand, the rapidity of glomerular disease onset in relation to administration of the COVID-19 vaccine seem to involve the participation of T-cells only as important mediators. T-cells react to foreign mRNA by inducing swift production of such cytokines (interferon γ, tumor necrosis factor α, interleukin 2) that could trigger podocytopathies and augment B-cell production of disease-specific antibodies in the susceptible patients. There are very few reports of vaccination-related thrombotic microangiopathy. Bitzan et al. reported five cases of TMA associated to influenza vaccination, suggesting that vaccines can activate complement directly even if the underlying pathogenic link remains to be clarified [13].

In our patient, the close temporal proximity (7 days) between the administration of the vaccine dose and SARS-CoV-2 positivity detection prevents us from drawing a causal inference of the histological findings with either the vaccine or the disease itself. Another hypothesis to be evaluated is that, given the strict timeframe between the two events, the combination of both may have triggered the renal disease, although this possibility has been never speculated in the literature. Alternatively, we cannot exclude that the mild clinical and histological features of renal TMA could result from a protective effect of the vaccine, which might have mitigated thromboinflammation, endothelial injury and COVID-induced complement activation.

## Figures and Tables

**Figure 1 pathogens-10-01045-f001:**
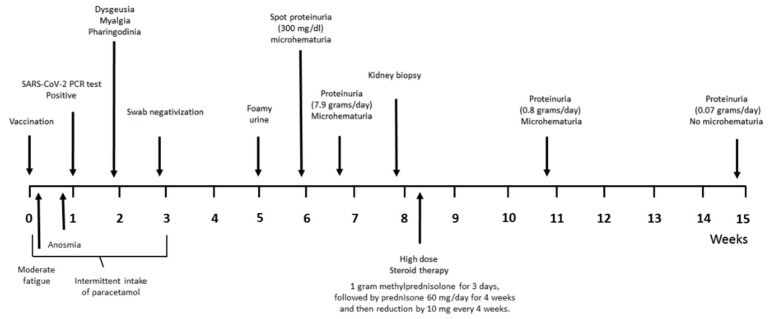
Timeline of clinical events.

**Figure 2 pathogens-10-01045-f002:**
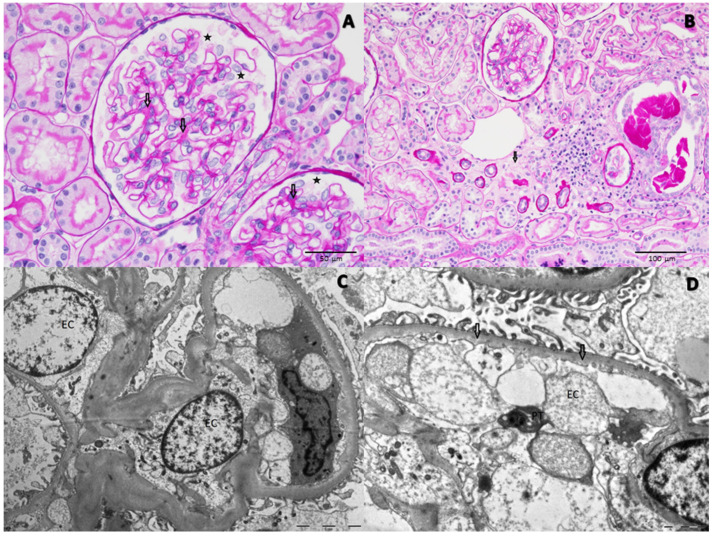
Light microscopy: (**A**) Minimal expansion of mesangial matrix (arrows), podocyte hypertrophy (stars) (PAS, 40X). (**B**) Focal tubular atrophy and interstitial fibrosis (arrow) (PAS 20X). Electron microscopy. (**C**) Hypertrophic and swollen cytoplasm of severely damaged endothelial cells (EC); the foot processes of podocytes seem to be regularly aligned and inter-digitated. (**D**) A very mild inner lamina rara widening (arrows) and loss of endothelial cell (EC) fenestrae. A platelet (PT) was also detected in the vessel lumen.

## Data Availability

The data presented in this study are available on request from the corresponding author.

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
