# Peer review of "Renal Thrombotic Microangiopathy in Concurrent COVID-19 Vaccination and Infection"

_pathogens, 2021, doi:10.3390/pathogens10081045_

Round 1

Reviewer 1 Report

Here, De Fabritiis et al report on the development nephrotic range proteinuria and hematuria in a 35 year old male patient 40 days after the first vaccination with a Covid-19 vaccine and 33 days after subsequent COVID-19 disease, confirmed by  positive PCR. This case report adds to a list of renal complications in both infection and vaccination related to SARS-CoV-2, which also includes thromboembolic events however it would present a first case of renal TMA. The clinical work-up is well presented, however, the individual´s profession is not of interest for the clinical presentation and should be omitted, not least for reasons of data protection.

Pictures from the patient´s renal biopsy show a histology in which the parenchyma appears, as the authors state, un-remarkable. In the electron micrographs one can observe some endothelial swelling indicating endothelial injury. The magnification and resolution of the electron micrographs does not allow to comment on the lamina rara widening, however, an area of foot process effacement is visible in panel C.

Although similar changes can be seen in early phases of thrombotic macroangiopathy, they can be seen also due to other causes, most notably other infections or forms of glomerulonephritis which should be excluded.  Neither timing of the occurrence of the renal disease after infection/vaccination, clinical parameters nor the results of histomorphological and ultrastructural investigations provided allow to draw the definite conclusion of TMA, not least because there seems an absence of thrombosis or fibrin aggregates or other indicators of TMA that have been described, such as fragmented erythrocytes or endothelial foam cells. In this context it would be interesting to see whether any fibrin can be detected using special stains by light microscopy or immunohistochemically in the FFPE sections. It is not clear how many glomeruli had been investigated by TEM, however the investigation of additional glomeruli may thus help to detect morphological evidence of TMA. If not, the authors should not draw the conclusion beyond the statement of endothelial injury in otherwise morphologically normal appearing glomeruli.

Of note, scale bars should be included in all images and, for clarity, arrows added to highlight changes observed.

Informed consent has been obtained from the patient, however, a statement on whether ethical approval for the study needs to be obtained, should be included.  

Author Response

Point 1: The clinical work-up is well presented, however, the individual´s profession is not of interest for the clinical presentation and should be omitted, not least for reasons of data protection.

Response 1: Thanks, we had disclosed the profession of the case only because that of GPs represents a high-risk category, but we agree with your concern about the privacy issue and we now removed it in the revised version.   

Point 2: Pictures from the patient´s renal biopsy show a histology in which the parenchyma appears, as the authors state, un-remarkable. In the electron micrographs one can observe some endothelial swelling indicating endothelial injury. The magnification and resolution of the electron micrographs does not allow to comment on the lamina rara widening, however, an area of foot process effacement is visible in panel C

Although similar changes can be seen in early phases of thrombotic macroangiopathy, they can be seen also due to other causes, most notably other infections or forms of glomerulonephritis which should be excluded.  Neither timing of the occurrence of the renal disease after infection/vaccination, clinical parameters nor the results of histomorphological and ultrastructural investigations provided allow to draw the definite conclusion of TMA, not least because there seems an absence of thrombosis or fibrin aggregates or other indicators of TMA that have been described, such as fragmented erythrocytes or endothelial foam cells. In this context it would be interesting to see whether any fibrin can be detected using special stains by light microscopy or immunohistochemically in the FFPE sections. It is not clear how many glomeruli had been investigated by TEM, however the investigation of additional glomeruli may thus help to detect morphological evidence of TMA. If not, the authors should not draw the conclusion beyond the statement of endothelial injury in otherwise morphologically normal appearing glomeruli.

Response 2: We apologize for the mistake describing “unremarkable” all the glomeruli examined here. Indeed, the “unremarkable” glomeruli are those studied by light microscopy, while electron microscopy was decisive to suggest a diagnosis by the presence of some ultrastructural alterations compatible with an initial phase of renal TMA. In summary, even if clinical, laboratory and histological result do not allow us to diagnose TMA with certainty, the ultrastructural features are suggestive of that.

Other renal diseases, such as focal and segmental glomerulosclerosis, post-infectious glomerulonephritis, membranoproliferative glomerulonephritis, can cause endothelial swelling and lamina rara widening, but the histological features of light microscopy and specific viral and bacterial tests ruled out these hypotheses.

The Figure 2 (C) has been changed  to better show the endothelial cell alterations (3 glomeruli were extensively studied).

Fibrin deposition was investigated by immunofluorescence on frozen sections and it was negative in all compartments. Unfortunately, we do not have available special stains by light microscopy or immunohistochemically in the FFPE sections.

These points have been addressed in the current version of the paper.

Point 3: Of note, scale bars should be included in all images and, for clarity, arrows added to highlight changes observed, as suggested by reviewer

Response 3: We have proceed inserted scale bars in all the images together with arrows and stars to highlight changes observed.

Point 4: Informed consent has been obtained from the patient, however, a statement on whether ethical approval for the study needs to be obtained, should be included.  

Response 4: We had already specified that in “Statements” section, specifically in the “Institutional Review Board Statement” (page 5, line 168-169). Anyhow, we have also added a sentence at the beginning of the case report description (paragraph 2 at page 2, lines 52-56 of the new version).

Reviewer 2 Report

This is an interesting case report that describes development of features of renal thrombotic microangiopathy following COVID-19 vaccination and disease. This topic is significant because given this ongoing pandemic with a relatively new virus and vaccine whose rarer adverse effects are still being characterized. The case report first briefly describes the virus and the vaccine and what is known about the virus’ renal effects. Then the case report of a single patient is described, including the patient’s timeline of getting the vaccine and the infection, symptoms after infection, histological findings, treatment, and recovery. The authors then explore previously reported renal effects of SARS-CoV-2 and how their work fits into and expands the literature. 

Major comments:

  1. There is no discussion of the patient’s symptoms or lack of symptoms in the seven days between when he was vaccinated and when he was PCR positive. Did the patient have any symptoms (subjective or measured) immediately after receipt of the vaccine, or in the time between receipt of the vaccine and positive test? I know that any symptoms in the time after vaccination may be confounded, since the patient may have already been infected asymptomatically with SARS-CoV-2 when he was vaccinated, but it is important to note the presence or absence in this period nonetheless, even with lots of caveats. On that note, it may help to make Figure 1 a bit more detailed: I would appreciate seeing more of the symptoms, and/or treatments, if those timeline are known.
  2. The authors posit that this renal pathology was due to the vaccine or due to the virus, but they do not mention or discuss that the close time frame means it could be due to both or to a combination of the vaccine and the virus. I believe some discussion along those lines would be beneficial to the paper.
  3. The article states that the Pfizer vaccine may have had a causative role in the patient’s renal disease. However, despite making this claim multiple times, there is no discussion of how the vaccine may have caused renal pathology. What biological mechanisms could be involved in vaccine-mediated, or vaccine-contributing, injury?
  4. Similarly, the discussion of the biology of how SARS-CoV-2 infection could have caused the renal injuries observed is cursory. I would like to see a more in-depth discussion of the biological mechanism that may be responsible.

Minor Comments:

  1. The PCR test is not a “COVID-19 test” but rather is a SARS-CoV-2 test. The PCR test detects the presence of the virus or the viral genetic material; however, a person can be positive for having SARS-CoV-2 genetic material while not having COVID-19. So the PCR test does not detect COVID-19, but rather detects SARS-CoV-2. Please adjust the wording in the report accordingly.
  2. Figure 1: Reading the words on the figure requires craning one’s neck. I would recommend making the words horizontal.
  3. The introduction should mention whether any renal adverse effects of the Pfizer vaccine are known, and if any are, what these are.
  4. Line 27: Seems it ought to be “does not allow us to associate…”
  5. Line 85: “0,07” should be “0.07”
  6. Line 117: I believe this should read: “were close not only…”
  7. Line 118: Capitalize “COVID”.

Author Response

Major comments:

Point 1: There is no discussion of the patient’s symptoms or lack of symptoms in the seven days between when he was vaccinated and when he was PCR positive. Did the patient have any symptoms (subjective or measured) immediately after receipt of the vaccine, or in the time between receipt of the vaccine and positive test? I know that any symptoms in the time after vaccination may be confounded, since the patient may have already been infected asymptomatically with SARS-CoV-2 when he was vaccinated, but it is important to note the presence or absence in this period nonetheless, even with lots of caveats. On that note, it may help to make Figure 1 a bit more detailed: I would appreciate seeing more of the symptoms, and/or treatments, if those timeline are known.

Response 1: The patient reported moderate fatigue the day after the vaccination and onset of anosmia the day before the SARS-CoV-2 PCR test was performed. During the first three weeks he signaled  intermittent intake of paracetamol. We cannot exclude anyhow that the proteinuria was already present before the patient noticed foamy urine. The steroid treatment was: 1 gram methylprednisolone for 3 days, followed by prednisone 60 mg/day for 4 weeks and then reduction by 10 mg every 4 weeks. All these details have been introduced in the amended Figure 1.

Point 2: The authors posit that this renal pathology was due to the vaccine or due to the virus, but they do not mention or discuss that the close timeframe means it could be due to both or to a combination of the vaccine and the virus. I believe some discussion along those lines would be beneficial to the paper.

Response 2: We gladly accept this suggestion, thus we added this hypothesis in the discussion (page 4, lines 164-167 of the new version).

Point 3: The article states that the Pfizer vaccine may have had a causative role in the patient’s renal disease. However, despite making this claim multiple times, there is no discussion of how the vaccine may have caused renal pathology. What biological mechanisms could be involved in vaccine-mediated, or vaccine-contributing, injury?

Response 3: The pathogenesis of vaccine-associated glomerular lesions has not been fully clarified for COVID-19 vaccines that induce the recipient’s cells to synthesize the COVID-19 spike protein. B-and T-cells are involved in effective immune response to the spike protein. On the other side, the rapidity of glomerular disease onset in relation to receipt of the COVID-19 vaccine implicates T-cells only as the most important mediators. T-cells react to foreign mRNA by inducing swift production of such cytokines (interferon γ, tumor necrosis factor α, interleukin 2) that could trigger podocytopathies and enhance B-cell production of disease-specific antibodies in the susceptible patients.

There are very few reports of vaccination-related thrombotic microangiopathy. Bitzan and Zieg reported 5 cases of TMA associated to influenza vaccination, suggesting that vaccine can directly activate complement cascade, even if the the pathomechanism linking TMA with influenza vaccines is still poorly understood (Bitzan M & Zieg J, Pediatr Nephrol 2018, doi: 10.1007/s00467-017-3783-4). In view of your valuable suggestion, we can added this comment in the discussion (page 4, lines 150-161 of the new version) with the related reference [13].

Point 4: Similarly, the discussion of the biology of how SARS-CoV-2 infection could have caused the renal injuries observed is cursory. I would like to see a more in-depth discussion of the biological mechanism that may be responsible.

Response 4: We have added more details regarding the biological mechanisms involved in COVID-related kidney injury. COVID-19 can infect endothelial cells owing to their high expression levels of angiotensin-converting enzyme 2 (ACE2) and also activate the complement system. In addition, complexes of COVID-19 specific antibodies and viral antigens might induce endothelial cell injury through activation of the C1 complex of the classical pathway and induction of antibody-dependent cytotoxicity (ADC). Pro-inflammatory cytokines and chemokines released by activated macrophages amplify the vicious cycle of vascular integrity disruption, vessel coagulation and thrombosis by degrading the endothelial glycocalyx, activating the coagulation system and dampening anticoagulant mechanisms. The adhesive phenotype of endothelial cells induced by inflammatory cytokines and chemokines promotes infiltration of neutrophils, which produce large amounts of histotoxic mediators, ultimately leading to injury of endothelial cells.

Minor Comments:

Point 1: The PCR test is not a “COVID-19 test” but rather is a SARS-CoV-2 test. The PCR test detects the presence of the virus or the viral genetic material; however, a person can be positive for having SARS-CoV-2 genetic material while not having COVID-19. So the PCR test does not detect COVID-19, but rather detects SARS-CoV-2. Please adjust the wording in the report accordingly.

Response 1: We have addressed this point in the revised version of the manuscript.

Point 2: Figure 1: Reading the words on the figure requires craning one’s neck. I would recommend making the words horizontal.

Response 2: We have corrected the figure accordingly.

Point 3: The introduction should mention whether any renal adverse effects of the Pfizer vaccine are known, and if any are, what these are.

Response 3: In line with available literature, we have detailed the known renal adverse effects of the mentioned vaccine (page 1, lines 38-41), namely: minimal change disease (Lebedev L et al, Am J Kidney Dis. 2021,  doi:10.1053/j.ajkd.2021.03.0102; D'Agati VD et al, Kidney Int. 2021, doi: 10.1016/j.kint.2021.04.035), relapse of minimal change disease (Komaba H et al, Am J Kidney Dis 2021, doi: 10.1053/j.ajkd.2021.05.006), IgA Nephropathy and anti-GBM glomerulonephritis (Tan HZ et al, Kidney Int 2021, doi: 10.1016/j.kint.2021.05.009), relapse of IgG4-related disease (Masset C et al, Kidney Int. 2021, doi: 10.1016/j.kint.2021.06.002). We have also added the related references (1-5).  

Point 4: Line 27: Seems it ought to be “does not allow us to associate…”

Response 4: This was a typo that we have now corrected.

Point 5: Line 85: “0,07” should be “0.07”

Response 5: We have also corrected this.

Point 6: Line 117: I believe this should read: “were close not only…”

Response 6: This was also a typo that we have now corrected.

Point 7: Line 118: Capitalize “COVID”.

Response 7: We have corrected this everywhere in the manuscript.

Round 2

Reviewer 1 Report

The changes to the manuscript have significantly improved it. However there are still concerns about some points raised previously. The authors have not mentioned, as requested, how many glomeruli were examined by TEM. Here, a sampling error may result in the failure to see fibrin precipitates. As there is a platelet trapped in a capillary loop, there may be more firm evidence of thrombotic microangiopathy. It may however also exclude evidence of more widespread, however focal foot process effacement.

With regard to confirmation of fibrin thrombi by light microscopy, there are several antibodies to fibrin available for use in Paraffin histology and should be applied to confirm the diagnosis. In the absence of direct signs of thrombosis, the term consistent with is therefore misleading and should be replaced with suggestive of, as the authors have mentioned in their responses, or simply microangiopathy.

Author Response

Second Round

Response to Reviewer 1 Comments

Point 1: The changes to the manuscript have significantly improved it. However there are still concerns about some points raised previously. The authors have not mentioned, as requested, how many glomeruli were examined by TEM. Here, a sampling error may result in the failure to see fibrin precipitates. As there is a platelet trapped in a capillary loop, there may be more firm evidence of thrombotic microangiopathy. It may however also exclude evidence of more widespread, however focal foot process effacement.

Response 1: Three glomeruli were deeply studied: fibrin precipitates were not detected; we specify this point in the new version of the paper (page 2, lines 85-86 and page 3 line 90-91)

Point 2: with regard to confirmation of fibrin thrombi by light microscopy, there are several antibodies to fibrin available for use in Paraffin histology and should be applied to confirm the diagnosis. In the absence of direct signs of thrombosis, the term consistent with is therefore misleading and should be replaced with suggestive of, as the authors have mentioned in their responses, or simply microangiopathy.

Response 2: Unfortunately, we currently do not have the possibility to equip ourselves with antibodies to fibrin. However, histological stains such as Masson's trichrome and PAS did not reveal fibrin deposition. In absence of direct signs of thrombosis, we replaced “consistent with” with “suggestive of”.

Reviewer 2 Report

This is an interesting case report that describes development of features of renal thrombotic microangiopathy following COVID-19 vaccination and disease. This topic is significant because given this ongoing pandemic with a relatively new virus and vaccine whose rarer adverse effects are still being characterized. The case report first briefly describes the virus and the vaccine and what is known about the virus’ renal effects. Then the case report of a single patient is described, including the patient’s timeline of getting the vaccine and the infection, symptoms after infection, histological findings, treatment, and recovery. The authors then explore previously reported renal effects of SARS-CoV-2 and how their work fits into and expands the literature. The authors have addressed all major and minor concerns from my initial review. There are a number of English language errors, but beyond this concern I have no additional comments to be addressed.

Minor comments:

English language errors are present in lines: 

  • 20: “male man” is redundant; “man” would be sufficient
  • 26, 153: the expression is not “on the other side” but rather “on the other hand”
  • 35: “Through the end of June 2021” is more correct than “Until the end of June 2021”
  • 119: rather than “manly”, should be “mainly”
  • 122: should be “the virus directly infects endothelial cells”
  • 133: “patients submitted to kidney biopsy” is unclear; perhaps “patients who underwent kidney biopsy” would be clearer, or some other adjustment
  • 139: should be “with no pre-existing conditions”

Author Response

Second Round

Response to Reviewer 2 Comments

Point 1: line 20: “male man” is redundant; “man” would be sufficient

Response 1: This was a typo that we have now corrected.

Point 2: line 26 and 153: the expression is not “on the other side” but rather “on the other hand”

Response 2: This point has been corrected.

Point 3: line 35: “Through the end of June 2021” is more correct than “Until the end of June 2021”

Response 3: This was a typo that we have now corrected.

Point 4: 119: rather than “manly”, should be “mainly”

Response 4: This was a typo that we have now corrected.

Point 5: line 122: should be “the virus directly infects endothelial cells”

Response 5: This was a typo that we have now corrected.

Point 6: line 133: “patients submitted to kidney biopsy” is unclear; perhaps “patients who underwent kidney biopsy” would be clearer, or some other adjustment

Response 6: This points has been corrected.

Point 7: line 139: should be “with no pre-existing conditions”

Response 7: This was a typo that we have now corrected.